# OtoMatch: Content-based eardrum image retrieval using deep learning

**Seda Camalan**[1]*, **Muhammad Khalid Khan Niazi**[1], **Aaron C. Moberly**[2], **Theodoros Teknos**[3], **Garth Essig**[2], **Charles Elmaraghy**[2], **Nazhat Taj-Schaal**[4], **Metin N. Gurcan**[1]

1 Center for Biomedical Informatics, Wake Forest School of Medicine, Winston-Salem, North Carolina, United States of America, 2 Department of Otolaryngology, Ohio State University, Columbus, Ohio, United States of America, 3 UH Seidman Cancer Center, Cleveland, Ohio, United States of America, 4 Department of Internal Medicine, Ohio State University, Columbus, Ohio, United States of America

* scamalan@wakehealth.edu

## Abstract

Acute infections of the middle ear are the most commonly treated childhood diseases. Because complications affect children's language learning and cognitive processes, it is essential to diagnose these diseases in a timely and accurate manner. The prevailing literature suggests that it is difficult to accurately diagnose these infections, even for experienced ear, nose, and throat (ENT) physicians. Advanced care practitioners (e.g., nurse practitioners, physician assistants) serve as first-line providers in many primary care settings and may benefit from additional guidance to appropriately determine the diagnosis and treatment of ear diseases. For this purpose, we designed a content-based image retrieval (CBIR) system (called OtoMatch) for normal, middle ear effusion, and tympanostomy tube conditions, operating on eardrum images captured with a digital otoscope. We present a method that enables the conversion of any convolutional neural network (trained for classification) into an image retrieval model. As a proof of concept, we converted a pre-trained deep learning model into an image retrieval system. We accomplished this by changing the fully connected layers into lookup tables. A database of 454 labeled eardrum images (179 normal, 179 effusion, and 96 tube cases) was used to train and test the system. On a 10-fold cross validation, the proposed method resulted in an average accuracy of 80.58% (SD 5.37%), and maximum F1 score of 0.90 while retrieving the most similar image from the database. These are promising results for the first study to demonstrate the feasibility of developing a CBIR system for eardrum images using the newly proposed methodology.

## Introduction

The financial burden of eardrum diseases to society is enormous; for example, more than $5 billion per year is spent on acute otitis media (OM) alone [1] because of unnecessary antibiotics and the over treatment of it. This contributes to antibiotic resistance as well. The incidence rates of acute OM and chronic OM are 10.9% and 4.8%, respectively, with 51.0% and 22.6% of these occurring in children under the age of five years [2]. Ear, nose, and throat (ENT)

**Data Availability Statement:** All eardrum image files are available from the Zenodo database (accession number(s) 10.5281/zenodo.3595567.).

**Funding:** The project described was supported in part by Award R21 DC016972 (PIs: Gurcan,

Moberly) from National Institute on Deafness and Other Communication Disorders. Moberly and Gurcan are directors of Otologic, Inc. The funders had no role in study design, data collection and analysis, decision to publish, or preparation of the manuscript.

**Competing interests:** Moberly and Gurcan are directors of Otologic, Inc. The other authors have declared that no competing interests exist. This does not alter our adherence to PLOS ONE policies on sharing data and materials.

physicians, emergency physicians, adult and pediatric primary care physicians and advanced care practitioners (e.g., nurse practitioners, physician assistants) in primary care settings, render the diagnosis of ear diseases. The diagnosis depends on technical skills as well as the accumulated clinical experience of the practitioners in viewing and diagnosing many different ear pathologies. To improve diagnostic accuracy and reduce the chances of using unnecessary antibiotics, previous computer-aided diagnosis studies have focused on the detection of OM [3–6]. In a recent study from our group [7], a decision fusion of predictions obtained from digital otoscopy images and tympanometry measurements has been applied for detecting eardrum abnormalities. Although the decision support provided by the tools described in those previous studies might provide some utility for many clinicians, providing a selection of similar-looking images–with already established diagnoses–would be helpful as an adjunct diagnostic support approach.

The question of image similarity has important applications in the medical domain because diagnostic decision-making has traditionally been used as evidence from patients' data (image and metadata) coupled with the physician's prior accumulated experiences of similar clinical cases [8]. Content-based image retrieval (CBIR) uses quantifiable features as the search criteria. Although several studies have used X-ray and cervicograph images for image retrieval [9–12], to the best of our knowledge, the advantages of this approach for otoscopic images have not yet been investigated. Unlike most existing medical image retrieval studies, our proposed system is based on deep learning techniques [13]. To date, there have been a few studies regarding the classification of otoscopic images using binary classifiers [14–16], which are deep neural network-based systems to classify images, but these have been limited to providing "normal" versus "abnormal" distinctions of the images [5, 17]. However, in these systems, similar images are not shown after the diagnostic decision.

There are some image-retrieval studies that use magnetic resonance images (MRI); however, they do not use deep learning techniques [18–21]. In contrast, deep learning-based image retrieval systems exist but for non-medical images [22–24]. Only a few studies utilize deep learning-based image retrieval systems for medical images, such as for retinal fundus images, brain MRI, and mammographic images [25–29].

To address the lack of decision support systems for eardrum diagnosis, we have developed a CBIR system for digital otoscope images, called *OtoMatch*. Our system has been developed for otoscope images and since it uses transfer learning it can potentially be generalized to other diseases. This is particularly useful when the number of images to train a network from scratch is limited. Here we present the first feasibility study of OtoMatch to retrieve otoscope images from patients with ear infections and normal cases. OtoMatch contains a preprocessing phase for the images, followed by feature extraction, lookup table generation, similarity measurements and image retrieval. The database and the preprocessing phase, which consists of removing black margin and time stamps, are described in section 2. This phase is followed by feature extraction using Inception-ResNet-v2, a pre-trained network [30]. Because the number of eardrum images was not sufficient to fully train a deep learning network, a pre-trained network is used with transfer learning [31] to extract features from three different layers as explained in Section 2.3. The methodology to convert features into the lookup table is described in Section 2.4. Similarity measurement and performance evaluation of the system are the subjects of Sections 2.4 and 2.5, respectively.

## Methodology

OtoMatch, our CBIR system, has five main phases (Fig 1): 1) preprocessing and data augmentation, 2) feature extraction, 3) lookup table generation, 4) similarity measurement, and 5)

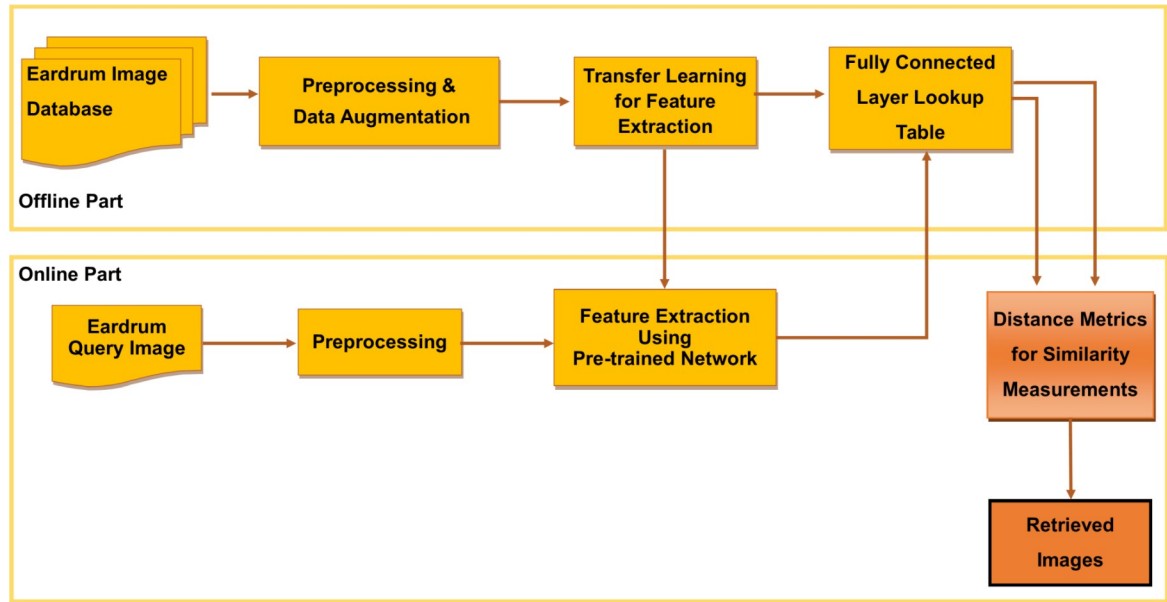

**Fig 1. Block diagram showing the workflow of the OtoMatch system.**

performance evaluation for image retrieval. This workflow is divided into two parts: offline and online. During the offline part, features from preprocessed images are converted into a lookup table. This conversion essentially changes a classification system into an image retrieval system. In the online part, the features of the query image are extracted using the same procedure to extract features from training images. Each part is described in detail in the following sub-sections.

During the offline part, a lookup table is generated, and a transfer learning system is developed for feature extraction. During the online phase, the same set of features is extracted from the query image and compared with the annotated set of cases in the database using different distance metrics to determine their similarity. The most similar images are then shown to the user.

## Database

The images used in this study were captured at Ear, Nose, and Throat (ENT) and primary care clinics from both adult and pediatric patients at the Ohio State University (OSU) and Nationwide Children's Hospital (NCH) in Columbus, Ohio, US with the IRB approval (Study Number: 2016H0011). Furthermore, all the samples were fully anonymized by the rules set by the Ohio State University, Institutional Review Board. Most of these high-resolution images were collected in size 1440 by 1080 pixels in the JPEG format. However, we also selected some frames from otoscopy video clips with the same size and resolution to increase the number of images in the tympanostomy tube category because the initial number of images in this category was 51, much fewer than the normal and effusion images. A total of 454 eardrum images were used to train and test the OtoMatch as categorized in Table 1. The data that support the findings of this study are openly available in Zenodo at http://doi.org/10.5281/zenodo.3595567.

Fig 2 shows examples of images (after preprocessing steps) from each category listed in Table 1 to demonstrate similarities and differences among different categories. Images from some categories may be hard to distinguish from those in another category for an untrained person.

Table 1. Number of images for each class of eardrum types.

| Category | Number of Images |
|---|---|
| Middle Ear Effusion | 179 |
| Normal | 179 |
| Tympanostomy Tube | 96 |
| Total | 454 |

It's challenging for providers with limited experience (e.g. those who are not ENTs) to confidently differentiate between different categories of images.

As can be seen in the images in Fig 2, the first row of images demonstrates more distinguishable characteristics than those in the second row in terms of color, shape and clear visibility of malleus of eardrum, which are important visual diagnostic characteristics of the eardrum abnormalities.

## Data augmentation and preprocessing

It was challenging to collect an equal number of images for each diagnostic category. Originally, we had more images in the normal (179 images) and effusion (179 images) categories than the tympanostomy tube (51 images) category, in which some frames from otoscopy video clips were selected with the same size and resolution to increase the number of images. To circumvent this class imbalance issue during training, a data augmentation approach was used. This approach includes reflecting images both horizontally and vertically, rotating them randomly, scaling them in the range of 0.7 and 2, shearing them both horizontally and vertically within a range of 0 to 45 degrees and translating them within a pixel range from -30 to 30 pixels in both horizontal and vertical directions. In each training batch, a random transformation of the images is used with data augmentation.

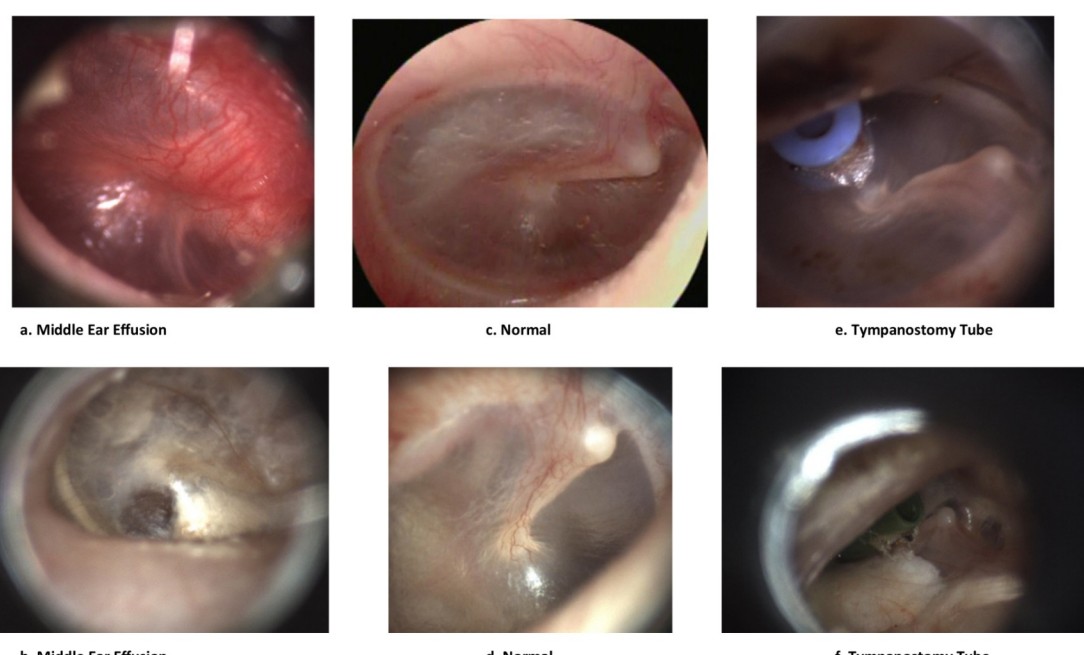

**Fig 2.** Example images of eardrum abnormalities: a & b- Middle Ear Effusion, c & d- Normal, e & f- Tympanostomy Tube.

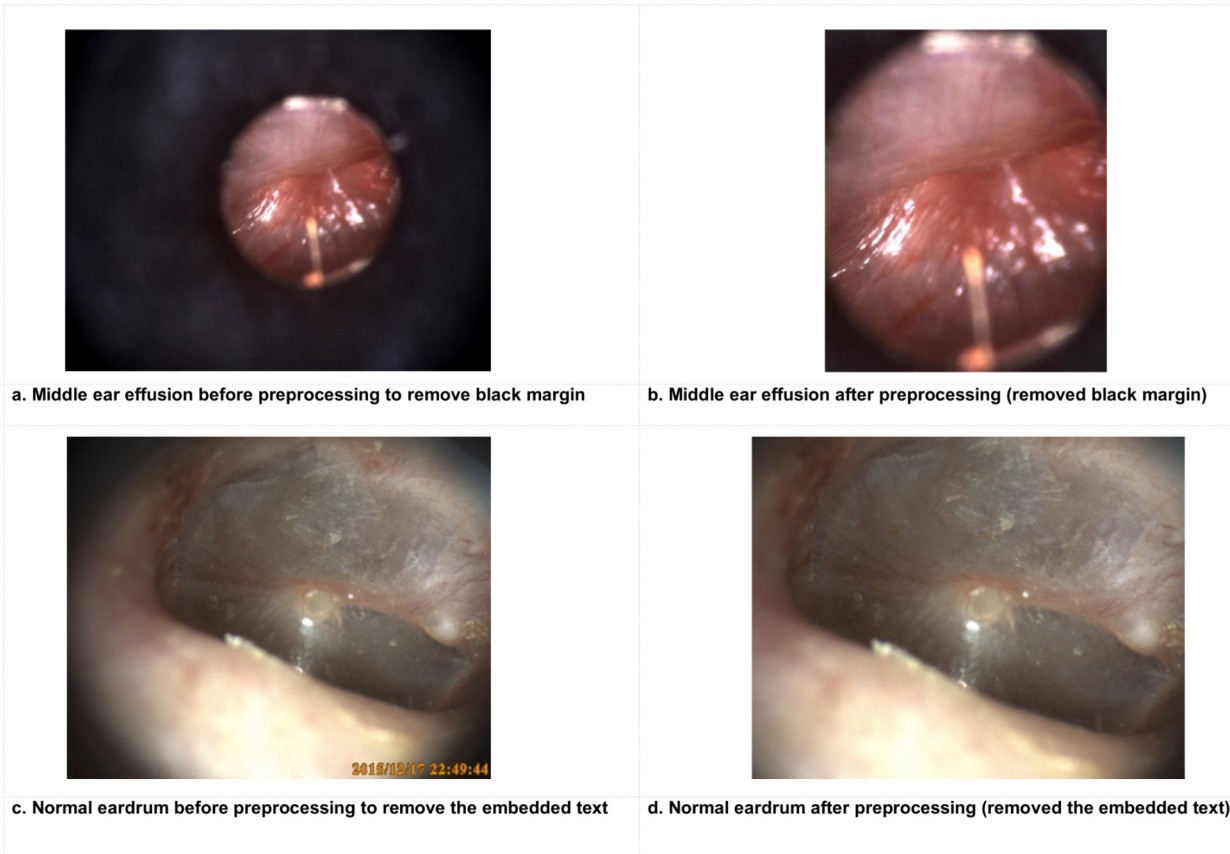

a. Middle ear effusion before preprocessing to remove black margin

b. Middle ear effusion after preprocessing (removed black margin)

c. Normal eardrum before preprocessing to remove the embedded text

d. Normal eardrum after preprocessing (removed the embedded text)

**Fig 3.** Example of before (a, c) and after (b, d) preprocessing to remove black margin and embedded time text.

Another challenge with otoscopy images is that only a portion of each image contains the eardrum. Therefore, before the data augmentation, the images were preprocessed to find the region of interest (ROI). Some images have large black margins and embedded text (to mark the capture date and time), which may negatively affect the values of the features and similarity measurements, hence needed to be removed or their impact minimized. To reduce the black margin around the eardrum, we first segmented the eardrum by converting it to a binary image and then finding its bounding box where the first white area started, and the last white area finished both in left to right and up to bottom. Anything beyond the bounding box was removed from consideration (i.e., those surrounding black regions). Because the time text was embedded automatically, always at the right bottom, these fixed location regions were excluded. Fig 3 demonstrates the image preprocessing steps for black boundary reduction and text removal.

## Feature extraction

Feature extraction is one of the most critical phases of OtoMatch that affects the similarity of the query image to the retrieved images. We propose a methodology to make use of a deep learning model to extract features for an image retrieval system. For this study, we showed how transfer learning can be converted into an image retrieval system. Although convolutional neural networks (CNNs) are extensively used for classification [32–35], their use in CBIR systems is rare [25–28]. In OtoMatch, we present a system that can be generalized to other

applications, especially useful when the number of cases in the database is not sufficient to fully train a deep learning system. We also demonstrate how to effectively use transfer learning within the CBIR context.

Transfer learning is a machine learning method that uses features learned from a problem in one domain and applies them to a problem in a different domain. For example, in Shin *et al.*'s work[36], AlexNet [37] and GoogLeNet [38], both previously trained on the ImageNet [37] dataset, are retrained to 1) analyze the condition of eardrum images if they are 'normal' or 'abnormal' and 2) detect and segment major structures of the eardrum [39]. In the past, there were a few attempts to categorize eardrum abnormalities into different classes [14, 15]. However, none of them applied CBIR for eardrum images. Classification of the eardrum abnormalities is not always sufficient for a decision support system because the end user may not fully understand the decision made by the computerized analysis. Presenting several examples of similar images may increase the confidence of the end user. For this reason, in this paper, we propose a method to retrieve similar images depending on their class of abnormality. The transfer learning is achieved by training the last few layers of Inception- ResNet-v2 neural network [30]. This method has the ability to acquire knowledge by automatically extracting features from eardrum images and using them to retrieve similar images to a query image through a lookup table.

For OtoMatch, we used Inception-ResNet-V2 Convolutional Neural Network (CNN), which was pre-trained and validated with 50,000 images set to classify 1,000 object categories and learned rich feature representations with 825 layers. Retraining the whole network requires a huge number of images. With our limited dataset, retraining the whole network is highly likely to result in overfitting [40]. For this reason, we opted to freeze the first 820 layers, limiting the number of parameters required to retrain the network. We retrained the last three layers (prediction, softmax, and classification) of the pre-trained network with otoscope images in our database. The choice of the number of retraining layers was mainly driven by the number of training images in our database and the number of parameters that we can learn from the database. Our feature extraction methodology only requires learning parameters within the last five layers of the Inception-ResNet-V2. After retraining the network, the resulting features were subjected to pooling that mapped each image into a vector of 1,536 features. Fig 4 illustrates the feature extraction phase.

The first 820 layers of the network (shown in blue) are frozen layers. The remaining five layers are retrained to extract features and classify the new set of images.

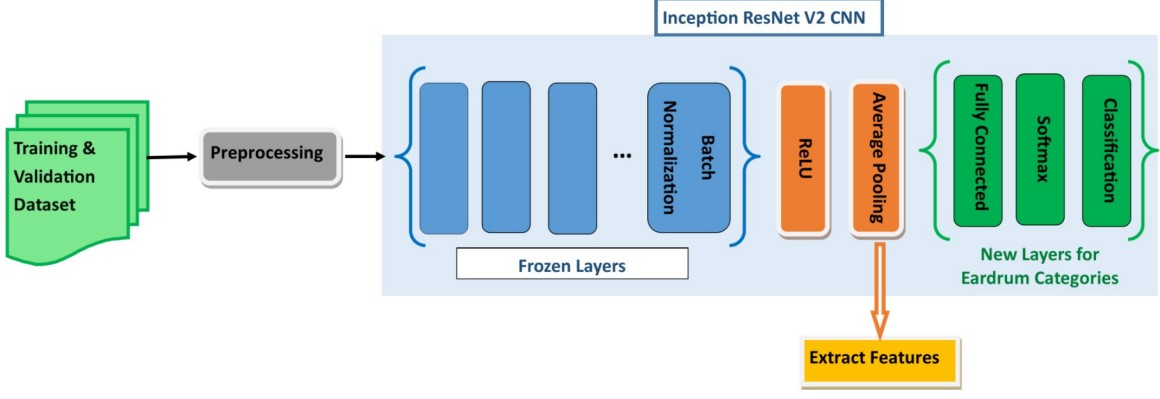

**Fig 4. Block diagram of feature extraction.**

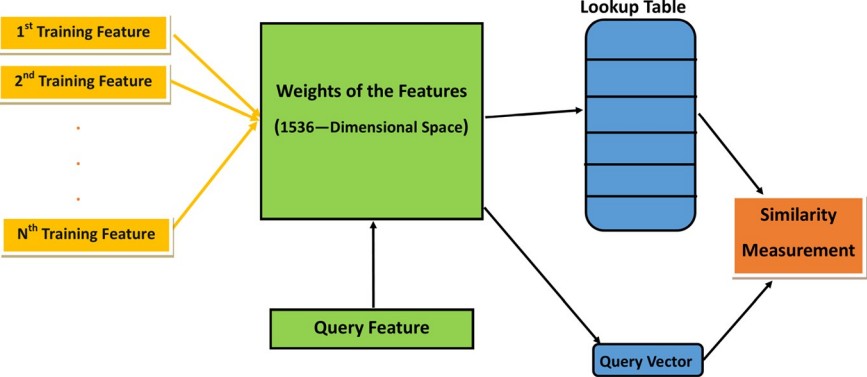

**Fig 5. Training features and query feature process before the similarity measurement.**

## Lookup table in fully connected layer

To retrieve the similar images from the training dataset, we need to compute the similarity or distance between two features of the images. For this purpose, we relied on the output of the fully connected layer, which produces 1 x 3 vectors for each training and test image, where three is the number of image categories in our database (normal, effusion, tympanostomy tube). The features are a 1536x1 vector produced as the output of the average pooling layer. Therefore, the weights are a 1536 x 3 matrix of the fully connected layer. When the transpose of the feature vector is multiplied by the weight vector, it produces a 1x 3 vector, which is established for each of the training set of images. When these vectors are turned to rows of a matrix (size of the number of training images x 3), this constitutes the lookup table. Any query image also goes through a similar process to create a 1 x 3 vector. In order to retrieve similar images to the query image, we use a similarity metric that is a function of the distance in the lookup table. This approach is demonstrated in Fig 5.

Intensity of learned features which may be defined as 1536-Dimensional space used to create lookup table.

From technical perspective, the fully connected layer enabled us to create a three-dimensional (3D) subspace for computing the image similarity between the test and training images. Each query (test) image is first mapped by projecting it into this 3D subspace, and the closest vector in this subspace is declared as the most similar image from the training set. Depending on the number of categories, the dimension of the subspace changes. The similarity function maps a feature vector to a real number, i.e. $R^n \rightarrow R$, where $n$ is the dimension of the feature vector, in our case, a 3 dimensional feature vector.

**Similarity measurements.** In order to find the most similar images, distance functions are used in CBIR systems. In OtoMatch, we used two similarity metrics: ChebyChev distance and Cosine distance. They are defined as:

$$\text{Chebychev} \qquad d_{st} = max_j \{|x_{sj} - y_{tj}|\} \qquad (1)$$

$$\text{Cosine} \qquad d_{st} = \left(1 - \frac{x_s y\prime_t}{\sqrt{(x_s x\prime_s)(y_t y\prime_t)}}\right) \qquad (2)$$

where $x$ and $y$ are the vectors whose distance are measured, $s$ and $t$ are the indices of the corresponding vectors, $d$ is the distance between these two vectors.

**Performance evaluation.** We evaluated our CBIR system in terms of precision, recall, and F1 score according to the number of query image and the retrieved images. Precision was calculated by dividing the number of true positives (positive samples in the test set that are correctly predicted) by the number of retrieved images across all query images. Recall was the rate of the true positives over the number of relevant images in the database. F1 score, which was used to define the accuracy, was calculated as

$$F1 = 2 * \frac{precision * recall}{precision + recall} \qquad (3)$$

Additionally, accuracy was used as an evaluation criterion of the system and defined as:

$$Accuracy = \frac{\sum True\ positive + \sum True\ negative}{\sum Total\ population} \qquad (4)$$

where true negatives are the negative samples in the test set that were correctly predicted.

## Results and discussion

Image retrieval systems using deep learning are not common, and our proposed system, Oto-Match, is a novel application of using deep learning for image retrieval within the context of eardrum abnormality diagnosis. Since deep learning is mostly used for classification, its use for CBIR systems is novel and can also be generalized other medical images and diseases. This proof of concept study showed that even with a limited amount of data, powerful deep learning frameworks can be used with the help of the transfer learning.

In our database (see Table 1), while effusion and normal categories were equal, the number of images in the tympanostomy tube category was less than the others, even after we added images from selected frames of the videos in the tympanostomy tube category. Because the learned model is biased towards the majority classes (i.e., those with higher number of images) to minimize the overall error rate, imbalanced datasets adversely impact the performance of the classifiers. As observed in the study [29], when the training set is balanced, the sensitivity and specificity gap significantly decreases. Therefore, we used random resampling methods to have an equal ratio of each class in training, validation and testing sets. Subsets of these images were used in CBIR training, validation and testing in order to have a balanced number of images from each class.

The database was divided into 10-folds randomly (k = 10 folds cross validation), and a similar number of images from each category were used in the training, validation and test sets. The training set was a sample of images used to train the system (i.e., to determine weights and biases in the Neural Network). For each fold, 10% of the database was randomly selected as test images, ensuring that we have similar number of images from each category in the test set. The rest of the data was divided into two groups: train (70%) and validation (30%). These training data were used for learning the network parameters. The validation set was used to minimize the effect of overfitting, i.e., fine tuning the hyperparameters of the network. Our trained network was pretrained for classification, and while training the network with our eardrum images, we also trained the system for classification of eardrum categories. Although image retrieval is different from classification because similarity of images is more indicative than deciding the category of the image, our system uses classification approach for image retrieval. In this respect, the accuracy results of each training fold according to the classification aspect are reported in Table 2.

For each fold, there were 40–44 (for different folds number of image changes) images used as the test set and 118–122 images used as the validation set. From test and validation sets,

**Table 2. Transfer learning validation and test accuracy values for each fold.**

| Fold | Validation Accuracy | Test Accuracy |
|---|---|---|
| 1 | 91.26% | 97.62% |
| 2 | 86.54% | 92.50% |
| 3 | 86.05% | 90.48% |
| 4 | 89.57% | 84.21% |
| 5 | 81.97% | 78.26% |
| 6 | 88.98% | 85.37% |
| 7 | 93.33% | 91.43% |
| 8 | 89.09% | 82.22% |
| 9 | 87.96% | 93.33% |
| 10 | 85.85% | 86.05% |
| Average | 88.06% | 88.15% |
| Standard Deviation | 2.87% | 5.33% |

images that were out of focus were discarded. Then the training and validation was carried out on each fold. For each fold, while training the network, validation accuracy was also recorded. After training phase, each fold test images were tested according to their classes on the trained network. The number of images in the test set are three times less than the number of images in the validation set. As a consequence, even when one of the images in the test set is misclassified, the accuracy of the test fold decreases by 2.5%; however, it only decreases by 0.8% in the validation set. For this reason, the results in the validation set are more consistent across folds than those in the test set. This can also be observed in the reported standard deviation values in Table 2. However, the average classification accuracies are comparable 88.06% (±2.87%) for the validation and 88.15% (±5.33%) for the test set. Please see page 14 line 276 for details. For retrieval, the images were first converted into features vectors. Then, these features were mapped into a 3D space using the fully connected layers. We use these 3D vectors to create a lookup table. During testing, each input image was first mapped into a 3D space and matched to the closest vector in the lookup table using two different similarity functions. To have consistent and robust results for each fold, each test image was queried for retrieving one image, three, five, seven and nine images at a time for each similarity function. The accuracy results of image retrieval system are shown in Table 3.

As can be observed in Table 3, average accuracy values while retrieving 1, 3, 5, 7, and 9 similar images are close to each other. For the Chebychev distance metrics, average accuracies are around 81% (±4%) while Cosine distance metrics average accuracies are around 82% (±4%). Again, these results are nearly identical.

For three of the folds, 3, 5, and 8, the image retrieval accuracies are less than 80%, which decreases the average accuracy with respect to other folds and unexpected according to the classification accuracy (reported in Table 2 has higher accuracy for these folds) of the transfer learning. However, when we consider the images in the test images (fold-3 has 43, fold-5 and fold-8 has 44 images), we observed that for some images, shown in Fig 6, it is challenging to find similar images in the training set because of the poor quality of the query image. These four images among 44 images are responsible for a drop of 9.09% accuracy. Selecting frames from videos or selecting the ROI in the image increase the accuracy of retrieval. Therefore, improving the selection of the ROI improves the performance.

We also explored the performance of OtoMatch according to the type of abnormality. The categorical F1-scores graphical representation is shown in Fig 7, where the maximum F1 score is 0.85, belonging to the tympanostomy tube category while retrieving a single image from the

**Table 3. Accuracy of OtoMatch while retrieving 1, 3, 5, 7, and 9 images for each 10-fold.**

| Fold | Retrieve 1 Image | | Retrieve 3 Images | | Retrieve 5 Images | | Retrieve 7 Images | | Retrieve 9 Images | |
|---|---|---|---|---|---|---|---|---|---|---|
| | Chebychev | Cosine | Chebychev | Cosine | Chebychev | Cosine | Chebychev | Cosine | Chebychev | Cosine |
| 1 | 84.09% | 79.55% | 86.36% | 85.61% | 85.45% | 85.00% | 86.69% | 85.06% | 87.12% | 84.85% |
| 2 | 87.50% | 87.50% | 86.67% | 90.00% | 87.00% | 89.50% | 87.14% | 88.93% | 87.22% | 88.33% |
| 3 | 74.42% | 69.77% | 78.29% | 78.29% | 79.53% | 80.00% | 80.40% | 80.73% | 79.59% | 80.88% |
| 4 | 84.21% | 86.84% | 83.33% | 85.09% | 82.63% | 85.26% | 84.59% | 84.59% | 83.63% | 81.87% |
| 5 | 77.27% | 72.73% | 74.24% | 75.00% | 73.18% | 74.55% | 74.03% | 72.40% | 72.22% | 73.74% |
| 6 | 80.49% | 73.17% | 78.86% | 81.30% | 80.49% | 81.46% | 80.84% | 80.84% | 80.76% | 80.49% |
| 7 | 83.78% | 91.89% | 81.98% | 82.88% | 80.54% | 82.16% | 81.47% | 81.85% | 80.48% | 81.98% |
| 8 | 75.00% | 88.64% | 77.27% | 77.27% | 75.00% | 78.18% | 74.68% | 79.55% | 75.00% | 79.80% |
| 9 | 80.95% | 83.33% | 81.75% | 82.54% | 82.38% | 83.81% | 82.65% | 84.69% | 82.80% | 84.92% |
| 10 | 83.72% | 83.72% | 86.82% | 85.27% | 85.58% | 86.05% | 85.38% | 86.05% | 84.75% | 85.79% |
| **Average** | **81.14%** | **81.71%** | **81.56%** | **82.33%** | **81.18%** | **82.60%** | **81.79%** | **82.47%** | **81.36%** | **82.26%** |
| **Standard Deviation** | **3.94%** | **6.87%** | **3.93%** | **4.08%** | **4.05%** | **3.89%** | **4.13%** | **4.12%** | **4.42%** | **3.66%** |

database. Although the number of images in the tube category is the lowest, the highest performance can be attributed to particularly distinguishing characteristics of tubes, which can be easier to identify than the subtle signs of abnormalities in the other two categories. On the other hand, middle ear effusion is the least accurate category because effusions present themselves in a wide spectrum of images.

We also observed that the performance of the system depends on the quality of the query image. For example, in Fig 8A, the query image is clear and well-focused, so the resulting accuracy is high. When the query image becomes blurry (e.g., Fig 8B), the performance decreases. In our future studies, we will develop a system to automatically grade the quality of images, so that we can also evaluate the dependence of the system performance on the quality of query images.

Currently, there is no similar or comparable system using CBIR for otitis media in the literature. Therefore, we compared the performance of our system with traditional K-Nearest Neighbor (KNN) and Support Vector Machine (SVM) classification method used with hand-crafted features. This consisted of HSV color space histogram given as $Hist(I_{HSV})$, color auto correlogram as $I_{CAC}$, the first two color moments from each RGB channel as $M(I_{RGB}^1)M(I_{RGB}^2)$ and Gabor Wavelet transform of the gray-scale version of the original color image as $GWT(I_{gray})$ [41]:

$$f_1 = [Hist(I_{HSV})I_{CAC}M(I_{RGB}^1)M(I_{RGB}^2)GWT(I_{gray})]$$

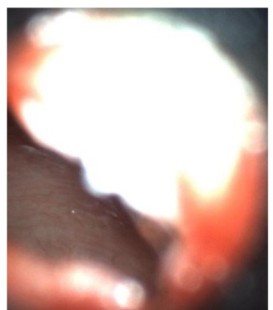 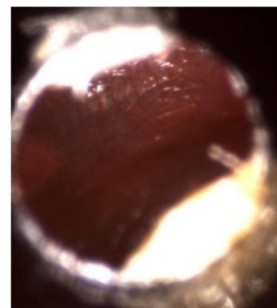 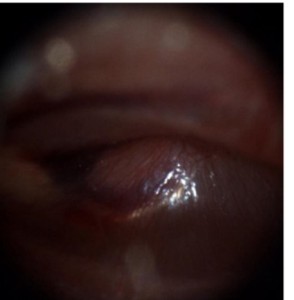 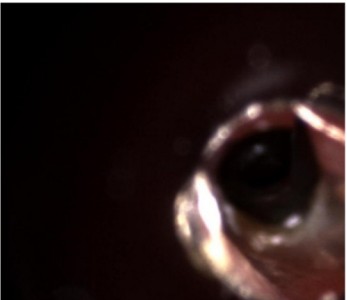

**Fig 6. Some difficult to determine diagnosis images from folds: 3, 5, and 8.**

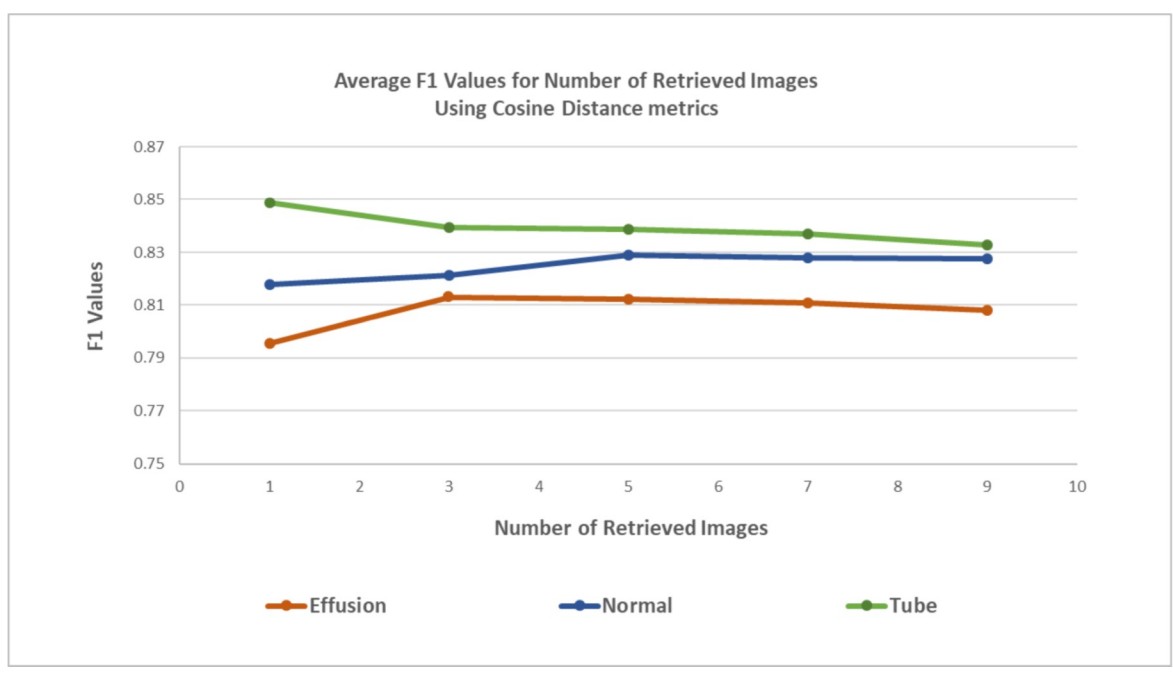

**Fig 7. F1 Values for each category for Cosine distance metric.**

Using the same dataset in these handcrafted and traditional classification methods, the results of the system are shown in Table 4 and Table 5.

Table **6** combines the averages and standard deviations of 10-fold results of CBIR systems OtoMatch, KNN and SVM classification with handcrafted features for 1, 3, 5, 7, and 9 retrieve images with Chebychev and Cosine similarity measurements.

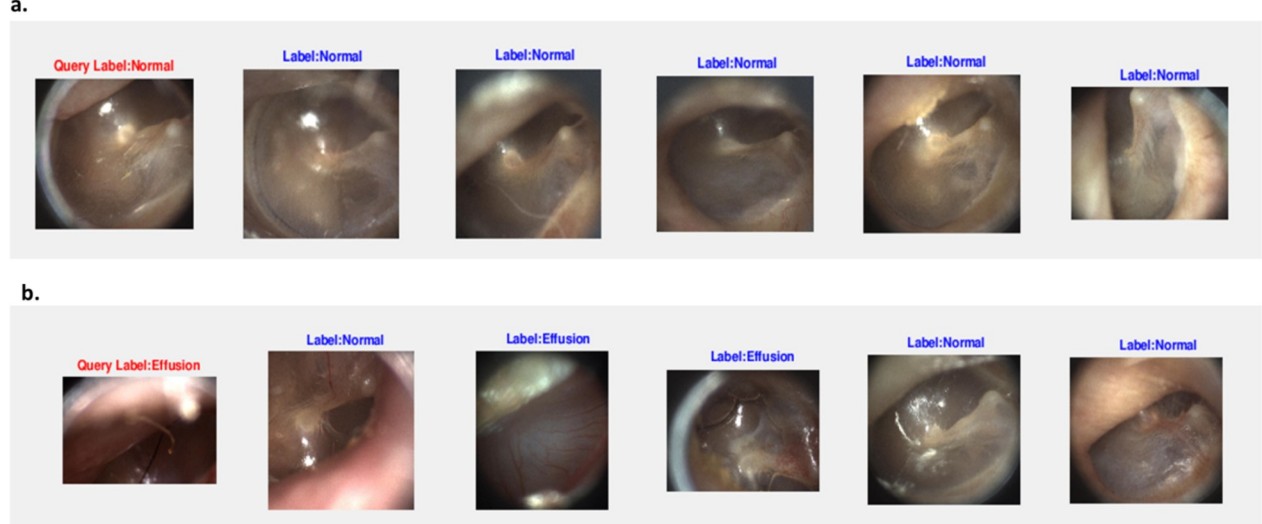

**Fig 8. Effect of quality of the query image.** a. For a clear and well-focused query image, the image retrieval accuracy is 100% for five retrieved images selected with the Cosine similarity metric. b. For an out-of-focus query image, the image retrieval accuracy is 40% for five retrieved images selected with the Cosine similarity metric.

**Table 4. Retrieval accuracy of 1, 3, 5, 7, and 9 CBIR image retrievals with traditional KNN classification and handcrafted features for each 10-fold.**

| Fold | Retrieve 1 Image | | Retrieve 3 Images | | Retrieve 5 Images | | Retrieve 7 Images | | Retrieve 9 Images | |
|---|---|---|---|---|---|---|---|---|---|---|
| | Chebychev | Cosine | Chebychev | Cosine | Chebychev | Cosine | Chebychev | Cosine | Chebychev | Cosine |
| 1 | 65.22% | 82.61% | 68.84% | 68.84% | 64.35% | 69.57% | 62.42% | 68.94% | 60.87% | 66.67% |
| 2 | 66.67% | 66.67% | 60.00% | 63.70% | 55.11% | 56.89% | 52.38% | 55.87% | 52.17% | 55.80% |
| 3 | 67.39% | 71.74% | 58.70% | 63.04% | 56.09% | 57.39% | 53.73% | 55.90% | 50.97% | 52.90% |
| 4 | 57.78% | 55.56% | 59.26% | 53.33% | 57.78% | 58.22% | 54.60% | 54.60% | 58.02% | 57.78% |
| 5 | 66.67% | 68.89% | 69.63% | 71.85% | 66.22% | 67.11% | 61.90% | 61.27% | 61.70% | 62.28% |
| 6 | 55.26% | 60.53% | 71.93% | 62.28% | 66.32% | 61.05% | 64.66% | 60.15% | 57.31% | 59.65% |
| 7 | 63.16% | 63.16% | 59.65% | 57.89% | 58.95% | 57.37% | 59.40% | 56.39% | 59.40% | 56.39% |
| 8 | 65.79% | 60.53% | 63.16% | 61.40% | 65.26% | 62.11% | 64.29% | 58.65% | 62.28% | 58.77% |
| 9 | 65.79% | 63.16% | 65.79% | 63.16% | 66.84% | 62.11% | 64.29% | 61.65% | 62.87% | 61.99% |
| 10 | 63.04% | 63.04% | 65.22% | 62.32% | 62.61% | 60.00% | 63.04% | 60.87% | 60.53% | 59.65% |
| Average | 63.68% | 65.59% | 64.22% | 62.78% | 61.95% | 61.18% | 60.07% | 59.43% | 58.61% | 59.19% |
| Standard Deviation | 3.68% | 6.81% | 4.35% | 4.63% | 4.11% | 3.87% | 4.31% | 3.81% | 3.72% | 3.50% |

**Table 5. Retrieval accuracy of 1, 3, 5, 7, and 9 CBIR image retrievals with traditional SVM classification and handcrafted features for each 10-fold.**

| Fold | Retrieve 1 Image | | Retrieve 3 Images | | Retrieve 5 Images | | Retrieve 7 Images | | Retrieve 9 Images | |
|---|---|---|---|---|---|---|---|---|---|---|
| | Chebychev | Cosine | Chebychev | Cosine | Chebychev | Cosine | Chebychev | Cosine | Chebychev | Cosine |
| 1 | 60.87% | 69.57% | 60.14% | 63.77% | 60.43% | 59.13% | 59.63% | 59.63% | 55.56% | 59.90% |
| 2 | 68.89% | 57.78% | 59.26% | 55.56% | 51.56% | 56.00% | 51.43% | 49.21% | 47.10% | 51.69% |
| 3 | 50.00% | 52.17% | 55.07% | 62.32% | 55.65% | 56.09% | 53.73% | 55.90% | 50.97% | 52.90% |
| 4 | 57.78% | 66.67% | 59.26% | 53.33% | 57.78% | 58.22% | 54.60% | 54.60% | 58.02% | 57.78% |
| 5 | 64.44% | 62.22% | 62.96% | 57.04% | 56.89% | 53.78% | 54.29% | 53.33% | 56.43% | 58.48% |
| 6 | 55.26% | 60.53% | 60.53% | 61.40% | 62.11% | 59.47% | 64.66% | 60.15% | 57.31% | 59.65% |
| 7 | 57.89% | 57.89% | 58.77% | 57.89% | 58.95% | 57.37% | 57.89% | 56.77% | 57.14% | 54.89% |
| 8 | 55.26% | 52.63% | 63.16% | 62.28% | 57.89% | 55.26% | 58.27% | 59.02% | 59.77% | 53.01% |
| 9 | 65.79% | 63.16% | 61.40% | 64.04% | 62.11% | 60.53% | 59.77% | 62.03% | 58.65% | 59.40% |
| 10 | 63.04% | 67.39% | 63.04% | 61.59% | 59.57% | 54.78% | 58.07% | 53.42% | 53.22% | 54.97% |
| Average | 59.92% | 61.00% | 60.36% | 59.92% | 58.29% | 57.06% | 57.23% | 56.41% | 55.42% | 56.27% |
| Standard Deviation | 5.19% | 5.40% | 2.25% | 3.34% | 2.87% | 2.02% | 3.45% | 3.53% | 3.54% | 2.83% |

**Table 6. Comparison of image retrieval methods.** Average accuracies and standard deviations of 1, 3, 5, 7, and 9 content based image retrievals for 10-fold with Oto-Match, traditional KNN, and SVM classifications using handcrafted features.

| Retrieve # Image | Similarity Measurement | OtoMatch | | KNN classification and handcrafted features | | SVM classification and handcrafted features | |
|---|---|---|---|---|---|---|---|
| | | Average | Standard Deviation | Average | Standard Deviation | Average | Standard Deviation |
| Retrieve1 | Chebychev | 81.14% | 3.94% | 63.68% | 3.68% | 59.92% | 5.19% |
| | Cosine | 81.71% | 6.87% | 65.59% | 6.81% | 61.00% | 5.40% |
| Retrieve3 | Chebychev | 81.56% | 3.93% | 64.22% | 4.35% | 60.36% | 2.25% |
| | Cosine | 82.33% | 4.08% | 62.78% | 4.63% | 59.92% | 3.34% |
| Retrieve5 | Chebychev | 81.18% | 4.05% | 61.95% | 4.11% | 58.29% | 2.87% |
| | Cosine | **82.60%** | **3.89%** | 61.18% | 3.87% | 57.06% | 2.02% |
| Retrieve7 | Chebychev | 81.79% | 4.13% | 60.07% | 4.31% | 57.23% | 3.45% |
| | Cosine | 82.47% | 4.12% | 59.43% | 3.81% | 56.41% | 3.53% |
| Retrieve9 | Chebychev | 81.36% | 4.42% | 58.61% | 3.72% | 55.42% | 3.54% |
| | Cosine | 82.26% | 3.66% | 59.19% | 3.50% | 56.27% | 2.83% |

For each fold, accuracies are relatively variable and low compared to the performance of OtoMatch, and the standard deviation is higher when the number of retrieved images increases. For the CBIR system where KNN was used as classification, the highest accuracy (65.59% ±6.81%) is much lower than the average OtoMatch accuracy (82% ±4%). This is also true for the CBIR system where SVM was used as classification where the average accuracy is 61.00% (±5.40%). These results show that traditional CBIR methods with handcrafted features are not as successful for this particular problem, and the variability (as measured by standard deviation) is much higher for these traditional methods.

## Conclusion

In this study, we propose a CBIR system for eardrum images called OtoMatch. OtoMatch is the first system developed to assist clinicians in ear disease diagnosis by retrieving the most similar images from a database of annotated cases. Previous studies have attempted to classify normal, abnormal, and suspected areas of eardrum perforation [39], but to the best of our knowledge, this is the first CBIR system for otoscopy imaging. More importantly, this study aims to propose a generic model and can be used in conjunction with any convolutional neural network. We proposed a methodology to convert a deep learning model into an image retrieval system. For proof of concept, we showed how transfer learning can be used in the context of otoscopy imaging retrieval.

OtoMatch uses deep learning-based approaches to extract the features and creates a lookup table to compare the similarity of query images with training images in multi-dimensional space. According to the experimental results, for the three categories with the highest numbers of images in our database, the maximum test accuracy of the system using transfer learning was 82.60%, and the maximum F1 score was 0.85 for the images from the tympanostomy tube category and while retrieving a single most similar image. When compared to traditional CBIR methods with handcrafted features, OtoMatch produces higher accuracy with low variability between the folds, hence making it more amenable to generalizability of its performance.

We explored the variability of performance depending on some of the experimental design factors. Our studies with OtoMatch did not indicate any change in the performance when the similarity metric or the number of retrieved images changes. In contrast, we observed that the performance would depend on the type of abnormality. Our future studies will determine if such dependence would continue if the number and variety of the images in the database increase.

Limitations of the study include a relatively small number of images in each category. Future studies will include larger number of training images for each category and also the number of categories of the types of eardrum diseases. We also observed that the performance of the system is affected by the quality of the query image (Fig 8A and 8B). In our future studies, we will develop a system to automatically grade the quality of images, so that we can also evaluate the dependence of system performance on the quality of query images.

## Author Contributions

**Conceptualization:** Muhammad Khalid Khan Niazi, Metin N. Gurcan.

**Data curation:** Seda Camalan, Aaron C. Moberly, Theodoros Teknos, Garth Essig, Charles Elmaraghy, Nazhat Taj-Schaal, Metin N. Gurcan.

**Formal analysis:** Seda Camalan, Muhammad Khalid Khan Niazi, Metin N. Gurcan.

**Funding acquisition:** Metin N. Gurcan.

**Investigation:** Muhammad Khalid Khan Niazi, Metin N. Gurcan.

**Methodology:** Seda Camalan, Muhammad Khalid Khan Niazi.

**Project administration:** Metin N. Gurcan.

**Resources:** Metin N. Gurcan.

**Software:** Seda Camalan.

**Supervision:** Metin N. Gurcan.

**Validation:** Seda Camalan, Muhammad Khalid Khan Niazi, Aaron C. Moberly, Theodoros Teknos, Garth Essig, Charles Elmaraghy, Nazhat Taj-Schaal, Metin N. Gurcan.

**Visualization:** Seda Camalan, Muhammad Khalid Khan Niazi, Metin N. Gurcan.

**Writing – original draft:** Seda Camalan.

**Writing – review & editing:** Seda Camalan, Muhammad Khalid Khan Niazi, Aaron C. Moberly, Theodoros Teknos, Garth Essig, Charles Elmaraghy, Nazhat Taj-Schaal, Metin N. Gurcan.

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
