## [Decision Letter · Decision Letter 0]

7 Apr 2020

PONE-D-20-00237

OtoMatch: Content-based Eardrum Image Retrieval using Deep Learning

PLOS ONE

Dear Dr. Camalan,

Thank you for submitting your manuscript to PLOS ONE. After careful consideration, we feel that it has merit but does not fully meet PLOS ONE’s publication criteria as it currently stands. Therefore, we invite you to submit a revised version of the manuscript that addresses the points raised during the review process.

We would appreciate receiving your revised manuscript by May 22 2020 11:59PM. To enhance the reproducibility of your results, we recommend that if applicable you deposit your laboratory protocols in protocols.io, where a protocol can be assigned its own identifier (DOI) such that it can be cited independently in the future. For instructions see: http://journals.plos.org/plosone/s/submission-guidelines#loc-laboratory-protocols

We look forward to receiving your revised manuscript.

Kind regards,

Tao Song

Academic Editor

PLOS ONE

Journal Requirements:

Reviewers' comments:

Reviewer's Responses to Questions

**Comments to the Author**

1. Is the manuscript technically sound, and do the data support the conclusions?

Reviewer #1: Yes

Reviewer #2: Yes

2. Has the statistical analysis been performed appropriately and rigorously? 

Reviewer #1: Yes

Reviewer #2: Yes

3. Have the authors made all data underlying the findings in their manuscript fully available?

Reviewer #1: Yes

Reviewer #2: Yes

4. Is the manuscript presented in an intelligible fashion and written in standard English?

Reviewer #1: Yes

Reviewer #2: Yes

5. Review Comments to the Author

Reviewer #1: In this paper, the author proposed a CBIR system for eardrum images which uses deep learning-based approaches to extract the features and creates a lookup table to compare the similarity of query images with training images in multi-dimensional space. The results of the new system are better than other methods.

However, there are some details that should be paid attention.

1. In table 2, the differences between validation accuracy and test accuracy are obvious and sometimes validation is higher and in the other case text accuracy is higher. It is better to explain.

2. The quality of some figures are very poor, especially figure 1, 4 and 5.

3. when comparing the results with traditional methods, all the results are in separate tables. It would be better if you add another table or figure to make it calear.

Reviewer #2: This paper provides a content-based image retrieval (CBIR) system (called OtoMatch) and presents a method that enables the conversion of any convolutional neural network into an image retrieval model. The questions raised in the introduction are properly answered in the conclusions. The results are efficient and promising.

My advice is to accept the paper.

6. PLOS authors have the option to publish the peer review history of their article (what does this mean?). If published, this will include your full peer review and any attached files.

Reviewer #1: No

Reviewer #2: No

---

## [Author Response · Author response to Decision Letter 0]

17 Apr 2020

Reviewer's Responses to Questions

We thank the editor and the reviewers for their thoughtful comments and constructive criticism. We have thoroughly responded to all the comments, and their questions have brought further clarity to the revised manuscript. 

5. Review Comments to the Author 

Reviewer #1: In this paper, the author proposed a CBIR system for eardrum images which uses deep learning-based approaches to extract the features and creates a lookup table to compare the similarity of query images with training images in multi-dimensional space. The results of the new system are better than other methods.

However, there are some details that should be paid attention.

1. In table 2, the differences between validation accuracy and test accuracy are obvious and sometimes validation is higher and in the other case text accuracy is higher. It is better to explain.

Answer:

The number of images in the test set are three times less than the number of images in the validation set. As a consequence, even when one of the images in the test set is misclassified, the accuracy of the test fold decreases by 2.5%; however, it only decreases by 0.8% in the validation set. For this reason, the results in the validation set are more consistent across folds than those in the test set. This can also be observed in the reported standard deviation values in Table 2. However, the average classification accuracies are comparable 88.06% (±2.87%) for the validation and 88.15% (±5.33%) for the test set. Please see page 14 line 277 for details. 

2. The quality of some figures are very poor, especially figure 1, 4 and 5.

Answer: As per reviewer suggestion, we have redrawn our previous Figure 1, 4, and 5, which were saved in lower quality by mistake. Please find them in uploaded figures.

3. when comparing the results with traditional methods, all the results are in separate tables. It would be better if you add another table or figure to make it calear.

Answer: 

This is a really good suggestion. We have combined the results from all methods in Table 6 to reflect this change. Please see page 21 line 350 for details.

Table 6. Comparison of image retrieval methods. Average accuracies and standard deviations of 1, 3, 5, 7, and 9 content based image retrievals for 10-fold with OtoMatch, traditional KNN, and SVM classifications using handcrafted features.

Retrieve # Image Similarity Measurement OtoMatch KNN classification and handcrafted features SVM classification and handcrafted features

 Average Standard Average Standard Average Standard

 Deviation Deviation Deviation

Retrieve1 Chebychev 81.14% 3.94% 63.68% 3.68% 59.92% 5.19%

 Cosine 81.71% 6.87% 65.59% 6.81% 61.00% 5.40%

Retrieve3 Chebychev 81.56% 3.93% 64.22% 4.35% 60.36% 2.25%

 Cosine 82.33% 4.08% 62.78% 4.63% 59.92% 3.34%

Retrieve5 Chebychev 81.18% 4.05% 61.95% 4.11% 58.29% 2.87%

 Cosine 82.60% 3.89% 61.18% 3.87% 57.06% 2.02%

Retrieve7 Chebychev 81.79% 4.13% 60.07% 4.31% 57.23% 3.45%

 Cosine 82.47% 4.12% 59.43% 3.81% 56.41% 3.53%

Retrieve9 Chebychev 81.36% 4.42% 58.61% 3.72% 55.42% 3.54%

 Cosine 82.26% 3.66% 59.19% 3.50% 56.27% 2.83%

Table 6 combines the averages and standard deviations of 10-fold results of CBIR systems OtoMatch, KNN and SVM classification with handcrafted features for 1, 3, 5, 7, and 9 retrieve images with Chebychev and Cosine similarity measurements.

---

## [Editor Report · Decision Letter 1]

22 Apr 2020

OtoMatch: Content-based Eardrum Image Retrieval using Deep Learning

PONE-D-20-00237R1

Dear Dr. Camalan,

We are pleased to inform you that your manuscript has been judged scientifically suitable for publication and will be formally accepted for publication once it complies with all outstanding technical requirements.

With kind regards,

Tao Song

Academic Editor

PLOS ONE

---

## [Editor Report · Acceptance letter]

29 Apr 2020

PONE-D-20-00237R1 

OtoMatch: Content-based Eardrum Image Retrieval using Deep Learning 

Dear Dr. Camalan:

I am pleased to inform you that your manuscript has been deemed suitable for publication in PLOS ONE. Congratulations! Your manuscript is now with our production department. 

With kind regards,

on behalf of

Dr. Tao Song 

Academic Editor

PLOS ONE